# Evaluating Synergistic Effects of Hyaluronic Acid, Human Umbilical Cord-Derived Mesenchymal Stem Cells, and Growth Hormones in Knee Osteoarthritis: A Multi-Arm Randomized Trial

**DOI:** 10.3390/biomedicines12102332

**Published:** 2024-10-14

**Authors:** Ismail Hadisoebroto Dilogo, Anissa Feby Canintika, Bernadus Riyan Hartanto, Jacub Pandelaki, Irsa Gagah Himantoko

**Affiliations:** 1Department of Orthopaedics and Traumatology, Faculty of Medicine, Universitas Indonesia, Dr. Cipto Mangunkusumo Hospital, Jakarta 10430, Indonesia; anissafeby29@gmail.com (A.F.C.); bernadusriyan@gmail.com (B.R.H.); 2Stem Cell Medical Technology Integrated Service Unit, Dr. Cipto Mangunkusumo Hospital, CMU 2 Building 5th Floor, Jl. Diponegoro 71, Jakarta 10430, Indonesia; 3Stem Cell and Tissue Engineering Research Center, Indonesia Medical Education and Research Institute, Faculty of Medicine, Universitas Indonesia, Jakarta 10430, Indonesia; 4Department of Radiology, Faculty of Medicine, Universitas Indonesia, Dr. Cipto Mangunkusumo Hospital, Jakarta 10430, Indonesia; jacubp@gmail.com; 5Faculty of Medicine, Universitas Indonesia, Dr. Cipto Mangunkusumo Hospital, Jakarta 10430, Indonesia; irsagagah@gmail.com

**Keywords:** knee osteoarthritis, mesenchymal stem cells, hyaluronic acid, growth hormones, cartilage regeneration

## Abstract

Background: Knee osteoarthritis (OA) significantly affects quality of life and imposes economic burdens due to its prevalence and the disability it causes. The efficacy of current treatments is limited to alleviating the symptoms, and they cannot be used for regenerative purposes. This study aims to evaluate the efficacy and safety of combining hyaluronic acid (HA), human umbilical cord-derived mesenchymal stem cells (hUC-MSCs), and synthetic human growth hormone (somatotropin) in the treatment of knee OA, assessing pain relief, functional improvement, and cartilage regeneration. Methods: A four-arm, double-blind randomized trial was conducted with 51 knees from 28 subjects aged ≥50 with primary knee OA. The treatments involved were HA alone, HA with hUC-MSCs, HA with somatotropin, and a combination of all three. Efficacy was measured through the International Knee Documentation Committee (IKDC) score, Western Ontario and McMaster Universities Osteoarthritis Index (WOMAC), and visual analog score (VAS), and MRI T2 mapping of cartilage was conducted on pre-implantation at the 6th and 12th month. Results: All treatment arms showed improvements in the VAS and WOMAC scores over 12 months, suggesting some pain relief and functional improvement. However, MRI T2 mapping showed no significant cartilage regeneration across the groups. Conclusions: While the combined use of HA, hUC-MSCs, and somatotropin improved symptoms of knee OA, it did not enhance cartilage regeneration significantly. This study highlights the potential of these combinations for symptom management but underscores the need for further research to optimize these therapies for regenerative outcomes.

## 1. Introduction

Knee osteoarthritis (OA), commonly known as degenerative knee joint disease, is related to the mechanical change and the manifestation of progressive disease due to gradual wearing out of the knee joint articular cartilage, affecting the bone. In osteoarthritis, breakdown of the articular or hyaline cartilage in weight-bearing synovial joints results in alteration of the bone. In the later stages, radiographic abnormalities are tangible and identifiable and may include joint space narrowing, the presence of bony outgrowths known as osteophytes, and subchondral cysts [1]. The adipose tissue adjacent to the synovium, called the infrapatellar fat pad (IFP), is also believed to play a role in OA pathogenesis through the secretion of inflammatory cytokines and multiple endocrine factors, resulting in increased IFP volume in OA patients compared to their healthy counterparts [2]. Knee OA imposes a significant burden on patients, caregivers, and the public owing to its prevalence, impact on quality of life, and economic implications. Patients with knee osteoarthritis often experience pain that can lead to lifestyle modifications [1]. This pain not only affects their daily activities, but also contributes to a decrease in their overall quality of life [3]. The burden is further exacerbated by the increasing prevalence of knee OA, attributed to factors such as the global rise in obesity and an aging population [4]. Additionally, knee OA is highlighted as a major cause of disability, especially among older individuals, placing a strain on healthcare resources [5].

Despite the burden imposed by knee OA, current treatment options have limitations. The management of knee OA often involves a combination of pharmacological and nonpharmacological interventions. However, the effectiveness of these treatments can vary, and some patients may not achieve adequate pain relief or functional improvement [6]. Furthermore, the increasing prevalence of knee OA poses a challenge to healthcare systems, necessitating the development of more cost-effective and sustainable management strategies.

Hyaluronic acid (HA) has been shown to be safe and effective for knee osteoarthritis, offering long-lasting improvement in clinical parameters. While some studies have found that combining hyaluronic acid with corticosteroids enhances pain control, other research indicates that hyaluronic acid alone is a viable option for long-term pain relief and improved function [7,8]. Aside from hyaluronic acid, synthetic human growth hormone has been a sought-after active ingredient for knee osteoarthritis. A clinical trial by Rahimzadeh et al. found that adding growth hormone (somatropin) to platelet-rich plasma could be effective. Additionally, a rabbit model study by Palmieri et al. indicated that combining growth hormone with hyaluronic acid resulted in better outcomes than hyaluronic acid alone, suggesting enhanced therapeutic potential [9,10]

Mesenchymal stem cells (MSCs) have gained popularity because of their ease of harvesting, safety, and potential to differentiate into cartilage tissue [10]. Clinical trials have demonstrated the safety and effectiveness of MSCs in promoting cartilage regeneration and alleviating the symptoms of knee OA [11]. Additionally, studies have shown that stem cell-derived extracellular vesicles (EVs) can be effective in treating joint injuries and OA, offering regenerative effects and therapeutic benefits [12]. Despite the promising outcomes reported in various studies, there are still limitations and challenges associated with stem cell therapy for OA. The heterogeneity of cell entities and concomitant procedures in clinical studies have led to unclear evidence regarding the efficacy of MSCs in knee OA [13]. While preclinical and clinical trials have shown initial evidence of efficacy and safety in using MSC therapies for knee OA, more robust randomized controlled trials are needed to establish the definitive efficacy and efficiency of these treatments [14].

## 2. Materials and Methods

This four-arm, double-blind, randomized study was performed between November 2019 and October 2023 at Cipto Mangunkusumo General Hospital, Jakarta, Indonesia. The randomizer software randomizer.org (accessed on 12 November 2019) was used to generate a randomization sequence before subject recruitment. Upon enrollment, sequential envelopes were opened to determine group assignments. Randomization and allocation processes were carried out in separate teams to make ensure a double-blind process. The subjects and the orthopedic surgeon providing the intervention were blinded to the intervention during the study. The subjects involved in this study were patients who were at least 50 years old with a confirmed diagnosis of primary OA in one or both knees, according to the American College of Rheumatology (ACR criteria). Osteoarthritis was defined based on history, clinical examination, and radiographic abnormalities consistent with Kellgren–Lawrence grades I–IV at least 6 months before the beginning of the study.

This study was registered at ClinicalTrial (reference NCT03800810). Subsequent to approval by the Institutional Review Board (KET-1149/UN2.FI/ETIK/2019), informed consent was obtained under the guidelines of the most recent version of the Helsinki Declaration. Knee OA was categorized according to the Kellgren and Lawrence criteria using standard knee standing anteroposterior and supine lateral radiographs. One radiologist performed image interpretation and staging separately. The inclusion and exclusion criteria are listed in Table 1.

### 2.1. MRI

An MRI examination of the knee was carried out using a GE Optima MR450W (wide bore) 1. 5 T (GE Healthcare, GE Healthcare, Waukesha, WI, USA). After scanning, we used CartiGram software, FuncTool 9.4.04b (GE Healthcare, Waukesha, WI, USA) to process the scans and create a final colored T2 map, following further analysis using Advance Workstation 4.6 (AW 4.6). The following parameters were used to obtain the T2 mapping sequence: 4 min acquisition time; 62.5 kHz receiver bandwidth; 4 mm slice thickness; 1.5 mm slice gap; coronal plane; 256 × 256 matrix; 16 × 16 cm FOV; TR of 1000 ms; TEs of 8.3, 16.6, 24.9, 33.2, 41.4, 49.7, 58, and 66.3 ms; and color range of 25–75 ms.

The first MRI was performed, and then further MRIs were performed after six months and twelve months of the intervention. A routine imaging technique that entailed scanning in the axial, coronal, and sagittal planes was applied to perform knee MRIs. Further, in the axial and sagittal sections, a special cartilage sequence was used, namely T1-weighted FS spoilt 3D gradient echo. These studies were performed at the anterior, middle, and posterior positions on each compartment. The average thickness was then calculated after that. The follow-up scans were obtained using the same MR sequences and anatomic location as for the baseline scans.

### 2.2. Multiple-Harvest Method [15]

In this study, hUC-MSCs were produced by the Stem Cell Medical Technology Integrated Service Unit, Cipto Mangunkusumo General Hospital. Umbilical cords were obtained from healthy, term birth donors who had undergone a disease screening process. The umbilical cord was dissected, and the umbilical vein and artery were extracted and chopped into tiny pieces. To keep the pieces moist and stop them from floating, they were individually placed on 24-well plates and submerged in a small amount of umbilical cord-derived conditioned medium. After that, the samples were incubated at 37 °C with 5% CO_2_. Every day, the growth and desiccation of the cells were observed. At 80–90% confluence, cultures were harvested, and more medium was provided as needed. The collected cells were cultivated once more to yield a large enough quantity for patient administration.

### 2.3. Treatment

Prior to receiving the injection, subjects were recruited and subsequently allocated to one of the four treatment groups based on a pre-randomized treatment allocation table. Patients allocated to group 1 received 20 mg/2 mL (1 syringe) of intra-articular hyaluronic acid (HA) (Suplasyn, Mylan, Galway, Ireland) per week for three weeks; group 2 received a combination of intra-articular human umbilical cord-derived mesenchymal stem cells (hUC-MSCs) with HA in the first week and HA only for the subsequent two weeks; group 3 received a combination of 5.83 mg/1 mL (1 syringe) somatotropin (Saizen, Merck, Bari, Italy) with HA in the first two weeks and HA only in the last weeks; and group 4 received a combination of hUC-MSCs, HA, and somatotropin in the first week, HA and somatotropin in the second week, and HA only in the third week.

The subjects diagnosed with knee osteoarthritis received three injections in our trial. During the initial session, participants received an intra-articular injection according to the allocated treatment arm. During the second and third weeks, the individuals received a 2 mL HA injection. The dosage was consistent with that utilized in a caprine model of osteoarthritis in a study conducted by Murphy et al. [16]. hUC-MSCs were obtained using the multiple-harvest explant approach and cultured in a xeno-free mix containing 10% platelet lysate prepared in-house, as detailed in our previous study [17].

The subjects were then monitored in the 1st and 3rd months, followed by assessments every 3 months for up to 1 year. The visual analog score (VAS), the Western Ontario and McMaster Universities Osteoarthritis Index (WOMAC), and the International Knee Documentation Committee (IKDC) score were among the outcome measurements. Prior to implantation, as well as six and twelve months after the procedure, T2 mapping of the knee cartilage was carried out. All subjects were included in the analysis.

### 2.4. Outcome

The primary outcome of this study is the efficacy of the intervention on clinical outcomes, as assessed based on VAS, WOMAC, and IKDC. The secondary outcomes of this study include evidence of cartilage thickening, as proven radiologically

### 2.5. Sample Size


$$n_{1}=n_{2}={\left(\frac{(Z_{\alpha }+Z_{\beta })S}{{\left(x_{1}-x_{2}\right)}^{2}}\right)}^{2}$$


*n* = Minimum sample in each group.*Z_α_* = 1.96 (from Z table at type 1 error of 5%);*Z_β_* = 0.84 (from Z table at 80% power);*S* = Standard deviation from previous study [18] → 13;*x*_1_ − *x*_2_ = Difference between mean values from previous study [18] → 12.2.Minimum sample size in each group = 8.9 ≈ 9 samples/group.

### 2.6. Data Analysis

Statistical analyses were conducted using IBM SPSS Statistics for Windows (IBM) version 26 [19]. The numerical variables (VAS, WOMAC, IKDC, T2 map) at baseline, and at the 6th and 12th months after implantation, were analyzed for normality using a Shapiro–Wilk test. Subsequently, the data were further analyzed using repeated measures ANOVA in a mixed model. If the repeated measures ANOVA showed a significant result, the data were further analyzed using a Bonferroni post hoc test to provide direct comparisons of each variable. The T2 map was further analyzed based on whether the value was within a normal range (40–60) or outside the range (below 40 or above 60) throughout the follow-up period for each arm. The proportion for each group that were within the range underwent hypothetical testing using an Independent Samples Kruskal–Wallis test with Pairwise comparisons.

## 3. Results

### 3.1. Characteristics of the Subjects

Twenty-eight subjects were recruited for this study, with a total of 51 knees (20 bilateral and 11 unilateral knees) [see Figure 1]. No subjects were excluded. Eight (28.6%) patients were male, with a mean age of 53.16 + 9.11 years. The subjects had a mean BMI of 24.83 + 2.63 kg/m^2^. This study involved 51 knees that were further divided into four groups: Arm 1 received HA, Arm 2 received a combination of MSC + HA, Arm 3 received Somatotropin + HA, and Arm 4 received MSC + Somatotropin + HA. All patients were categorized as having a Kellgren–Lawrence Grade of I-II (100%) (Table 2).

### 3.2. Clinical Outcomes

Table 3 and Table 4 show the clinical outcomes assessed from the subjects, consisting of pain and functional assessments using VAS and WOMAC. Throughout 12 months of observation, all subjects, regardless of treatment allocation, showed a decreasing VAS, although the changes were statistically insignificant (*p* = 0.139). The WOMAC score also showed a similar trend of decreasing throughout the observation period for all groups; nevertheless, the changes were not statistically significant (*p* = 0.587). Therefore, the changes in the VAS and WOMAC scores notably decreased over 12 months, although these changes did not differ statistically.

### 3.3. Radiographic Outcome

T2 quantitative mapping of the MRIs was utilized to assess cartilage changes following intra-articular injection. The results were taken from baseline, 6 months, and 12 months for bilateral knees for both the medial and lateral parts of the knees. Repeated measures ANOVA tests were used to observe any significant changes found across groups and over time. The results showed no significant differences, and there was no particular trend over 12 months of observation for each treatment arm. The T2 mapping data between the lateral and medial of the left and right knees showed no significant difference across the treatment arms and over time, with *p*-values of 0.826, 0.802, 0.353, and 0.395 for the lateral and medial sides of the right knees, and lateral and medial sides of the left knees, respectively. The T2 map value for each timepoint and arm were also categorized into whether the value was within the target range of normal T2 map values (40–60) or whether it was outside of this range (below 40 or above 60). The analysis on both arms and timepoints on the T2 map classification showed significant results for the T2 map value of the medial side in the 12th month among the four arms (*p* = 0.024).

## 4. Discussion

The current treatment options for knee osteoarthritis encompass a range of approaches aimed at managing pain, improving function, and addressing the underlying joint pathology. Conventional management strategies focus on pain relief through joint-specific exercises and pharmacological interventions [20]. These interventions aim to alleviate symptoms, enhance mobility, and improve the overall quality of life for individuals with knee OA. Intra-articular injections, including corticosteroids and hyaluronic acid, are commonly used to provide localized relief from pain and inflammation in knee OA. These injections can help reduce symptoms and improve joint function, particularly in individuals who may not be suitable candidates for surgery or who wish to delay surgical intervention [21].

The literature on the effect of HA for knee OA suggests that intra-articular injections of HA can be beneficial in managing symptoms and improving joint function in patients with knee OA, as they can provide effective pain relief, reduce joint stiffness, and enhance physical function [20]. HA injections are considered an important nonsurgical treatment option for knee OA, along with other interventions such as corticosteroids, in providing long-term pain relief and improving joint function in knee OA, possibly through their anti-inflammatory, anabolic, and chondroprotective properties [7,22,23,24].

Several studies have been conducted on the effect of intra-articular injections of synthetic human growth hormone on knee OA, which suggest promising outcomes in improving knee joint function and reducing symptoms. A comparative double-blind clinical trial by Rahimzadeh demonstrated that adding growth hormone to platelet-rich plasma for intra-articular injection improved the function of the osteoarthritic knee joint in a short period of time, with no observed complications, indicating the beneficial effect of the combination of growth hormone and platelet-rich plasma for individuals with knee osteoarthritis [9]. Kim et al. also conducted a study on a rabbit model of collagenase-induced osteoarthritis and found that the intra-articular injection of growth hormone, in combination with hyaluronic acid, induced morphoangiogenesis and led to the formation of capillaries with unique characteristics, potentially contributing to joint repair and regeneration [25]. These findings suggest that synthetic human growth hormone may have additive effects when combined with other intra-articular treatments for osteoarthritis.

There are some reasons why cartilage regeneration does not happen. One possible reason could be inadvertent intravascular injection of hyaluronic acid, leading to rare but significant complications such as cutaneous necrosis. The repeated use of intra-articular injections, such as corticosteroids, may result in accelerated cartilage loss, potentially impeding the regenerative process. Moreover, the concentration and molecular weight of hyaluronan in synovial fluid are reduced in osteoarthritis, affecting joint lubrication and potentially impacting cartilage regeneration. Another factor to consider is the age of the patient, as older individuals may respond differently to treatments like platelet-rich plasma or hyaluronic acid injections, which could influence the regenerative outcomes [26].

Furthermore, the choice of substances injected, such as hypertonic dextrose, morrhuate sodium, or platelet-rich plasma, can stimulate growth factor and cytokine production, potentially influencing the regenerative capacity of the cartilage. Additionally, the accuracy of intra-articular injections is crucial to ensure that medications are delivered directly into the joint space, maximizing the therapeutic benefits and minimizing complications that could impede cartilage regeneration [27,28].

MSC administration for knee OA shows promising outcomes in improving knee joint function and reducing symptoms. We conducted a previous study on the impact of MSC administration on varying degrees of knee OA, showing better improvement in IKDC and WOMAC without any improvement in T2 MRI mapping [18]. Its effectiveness and safety for knee OA treatment have been explored and summarized in a systematic review and meta-analysis, indicating that MSCs have the potential to be beneficial in managing knee OA [29]. Additionally, a Phase IIb, randomized, placebo-controlled clinical trial was conducted on the intra-articular injection of autologous adipose tissue-derived MSCs for knee OA, reporting positive results in improving knee joint function [10].

This is a novel study on the administration of combined compounds that are deemed effective and safe in the literature. When the three compounds were combined, it was expected to show improved results in terms of pain level, functional scores, and cartilage thickness observed from T2 mapping. However, only the VAS and WOMAC scores showed a declining result over 12 months. These results aligned with a previous study on the impact of MSC administration on varying knee OA degrees [18]; however, the effects of the combined agents on T2 mapping remained insignificant. This study is restricted by the relatively limited sample size and short duration of the follow-up. Further research should be conducted to increase the number of subjects and study time, and/or investigate the alteration of the dosage or concomitant use of other compounds in combination with MSCs. Additionally, the use of other radiological modalities can be considered to examine the articular cartilage regeneration of the knee.

## 5. Conclusions

Our findings demonstrate that while treatments such as hyaluronic acid, mesenchymal stem cells, and growth hormones individually show promise in clinical settings, their combined effects do not significantly enhance cartilage regeneration, as measured by T2 MRI mapping. The clinical outcomes, primarily assessed through VAS and WOMAC scores, indicated a trend towards symptom relief across all treatment groups. Future research should focus on optimizing these therapies, possibly through novel combinations or enhanced delivery mechanisms, to improve their efficacy in cartilage regeneration and overall joint health. This study underscores the complexity of treating knee osteoarthritis and highlights the need for continued innovation in non-surgical interventions.

## Figures and Tables

**Figure 1 biomedicines-12-02332-f001:**
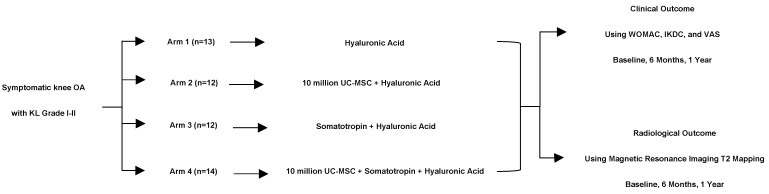
Subject flowchart.

**Table 1 biomedicines-12-02332-t001:** Inclusion and exclusion criteria.

Inclusion Criteria	Exclusion Criteria
Primary osteoarthritis in one or both knees	Experienced a knee joint infection or an infection in the skin or soft tissue around the knee
Radiographic evidence of Kellgren–Lawrence Grade I-II (mild to moderate) knee joint changes	History of surgery or other medical problem of the same knee (e.g., rheumatoid disease, gout, malalignment)
Body mass index (BMI) of below 30 kg/m^2^	Had a history of cancer
	History of hyaluronic or steroid injection in the past 6 months

**Table 2 biomedicines-12-02332-t002:** The characteristics of the subjects.

Variables	Total	Arm 1 (Hyaluronic Acid)	Arm 2 (MSC + Hyaluronic Acid)	Arm 3 (Somatotropin + Hyaluronic Acid)	Arm 4 (MSC + Somatotropin + Hyaluronic	*p* Value
No. of patients						
Patients	28					
Samples	51	13	12	12	14	
Gender						
Male	8 (28.6%)	3	7	1	2	0.022
Female	20 (71.4%)	10	5	11	12	
Age (years)	53.16 ± 9.11	48.75 ± 6.63	48.29 ± 9.12	56.1 ± 7.03	59.29 ± 10.23	0.03
BMI (kg/m^2^)	24.83 ± 2.63	25.58 ± 2.38	25.22 ± 4.57	27.02 ± 2.33	26.51 ± 3.67	0.074

**Table 3 biomedicines-12-02332-t003:** Outcomes of the mild osteoarthritis group at baseline, 6-month follow-up, 12-month follow-up.

Outcomes	Time	Mean (SD)				*p*-Value
		Arm 1	Arm 2	Arm 3	Arm 4	
VAS	Baseline	4.38 ± 0.74	5.71 ± 1.38	4.89 ± 1.17	5.71 ± 1.7	0.139
	6th month	3.13 ± 0.35	3 ± 0.58	3.33 ± 0.71	3.57 ± 0.98	
	12th month	2.13 ± 0.35	2	2.56 ± 1.01	2.71 ± 0.76	
WOMAC	Baseline	34.62 ± 5.7	30.79 ± 7.46	35.41 ± 10.6	34.96 ± 13.43	0.587
	6th month	21.54 ± 4.88	24.31 ± 4.79	25.17 ± 6.44	23.94 ± 6.89	
	12th month	12.22 ± 3.73	15.15 ± 1.7	18.04 ± 4.23	23.55 ± 20.73	
IKDC	Baseline	45.52 ± 8.7	38.8 ± 12	40.85 ± 9.35	44.82 ± 10.64	0.349
	6th month	58. 4 ± 6.58	47.5 ± 13.11	55.36 ± 6.95	52.59 ± 9.32	
	12th month	70.05 ± 13.1	48.77 ± 16.45	62.51 ± 9.96	58 ± 10.3	
T2 Map						
Medial	Baseline	59 ± 25. 44	49.42 ± 5	47.8 ± 9.12	68.24 ± 27.67	0.564
	6th month	48.77 ± 11.73	43.81 ± 4.52	43.67 ± 7.04	68.87 ± 39.8	
	12th month	135.79 ± 125.28	137.53 ± 117.25	199.22 ± 115.56	108.77 ± 107.40	
Lateral	Baseline	47.68 ± 9.64	50.24 ± 14.51	50.51 ± 10.62	53.33 ± 12.1	0.483
	6th month	52.71 ± 10.59	46.73 ± 2.40	68.04 ± 29.72	50.55 ± 8.05	
	12th month	43.46 ± 2.24	47.06 ± 5.72	45.62 ± 0.78	50.43 ± 11.89	

Hypothesis test using Mixed Model ANOVA and Bonferroni post hoc test.

**Table 4 biomedicines-12-02332-t004:** T2 Map at baseline, 6-month follow-up, 12-month follow-up.

Time	Variable	Target		*p*-Value
		Within	Off	
T2 map baseline medial	Arm 1	3 (37.5)	5 (62.5)	0.386
	Arm 2	5 (71.4)	2 (28.6)	
	Arm 3	3 (33.3)	6 (66.7)	
	Arm 4	2 (28.6)	5 (71.4)	
T2 map 6th month medial	Arm 1	2 (25)	6 (75)	0.314
	Arm 2	4 (57.1)	3 (42.9)	
	Arm 3	2 (22.2)	7 (77.8)	
	Arm 4	4 (57.1)	3 (42.9)	
T2 map 12th month medial	Arm 1	2 (25)	6 (75)	0.024 *
	Arm 2	3 (42.9)	4 (57.1)	
	Arm 3	0 (0)	9 (100)	
	Arm 4	5 (71.4)	2 (28.6)	
T2 map Baseline lateral	Arm 1	4 (50)	4 (50)	0.281
	Arm 2	3 (42.9)	4 (57.1)	
	Arm 3	2 (22.2)	7 (77.8)	
	Arm 4	5 (71.4)	2 (28.6)	
T2 map 6th month lateral	Arm 1	4 (50)	4 (50)	0.067
	Arm 2	5 (71.4)	2 (28.6)	
	Arm 3	2 (22.2)	7 (77.8)	
	Arm 4	6 (85.7)	1 (14.3)	
T2 map 12th month lateral	Arm 1	4 (50)	4 (50)	0.361
	Arm 2	5 (71.4)	2 (28.6)	
	Arm 3	3 (33.3)	6 (66.7)	
	Arm 4	17 (54.8)	1 (28.6)	

(*) Hypothesis test using Independent Samples Kruskal–Wallis test with Pairwise comparisons.

## Data Availability

The data presented in this study are available from the corresponding author upon request.

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
