# Peer review of "Evaluating Synergistic Effects of Hyaluronic Acid, Human Umbilical Cord-Derived Mesenchymal Stem Cells, and Growth Hormones in Knee Osteoarthritis: A Multi-Arm Randomized Trial"

_biomedicines, 2024, doi:10.3390/biomedicines12102332_

Round 1

Reviewer 1 Report (Previous Reviewer 2)

Comments and Suggestions for Authors

the paper is more realistic but it adds a little to what already known

Author Response

Thank you for your previous suggestions

The paper is significant as it contributes to knee osteoarthritis (OA) research by exploring a novel therapy combining hyaluronic acid, human umbilical cord-derived mesenchymal stem cells, and growth hormone. It addresses the limited efficacy of current knee OA treatments and the need for better non-surgical options. The paper presents a double-blind, randomized trial evaluating the therapy's efficacy and safety, showing improvements in pain and function but not in cartilage regeneration. It highlights the promise and challenges of using mesenchymal stem cells and growth factors in knee OA treatment, aligning with some previous studies. The findings emphasize the complexity of treating knee OA and the necessity for ongoing innovation and robust clinical trials.

Reviewer 2 Report (Previous Reviewer 1)

Comments and Suggestions for Authors

Issues and suggestions raised by the reviewer have been addressed properly

Author Response

Thank you for your previous suggestions

Reviewer 3 Report (New Reviewer)

Comments and Suggestions for Authors

The authors of the manuscript, 'Evaluating synergistic effects of hyaluronic acid, human umbilical cord-derived mesenchymal stem cells and growth hormones in knee osteoarthritis: A multiarm randomized trials' have tried to examine the collaborative role of three important therapeutic stuff for Knee OA. The paper is well-written and largely without any mistakes, typos or errors. The authors should ponder some of the following points and edit their manuscript for better presentation.

1.      The full form of IRB must be given for clarity (line 96), as in some institutions, it is called IEC (Institutional Ethical Committee), and in some, it is called ERB (Ethics Review Board) or REB (Research ethics board).

2.      The authors have written in the next line, ‘informed consent was sought……Declaration’. It should be,’ informed consent was taken. The authors are suggested to change it accordingly.

3.      Section 2.3 'Multiple Harvest Method' should be placed before section 2.2 'Treatment'.

4.      Number of patients must be mentioned in the section 2. ‘Materials and Methods’ along with the information, how many subjects had bilateral knee OA.

5.      Patients doing regular exercise must be taken in the exclusion criteria, as exercise significantly improves VAS and WOMAC scores.

6.      In the 4th paragraph (lines 262-271), the authors discussed possible scenarios of why cartilage was not regenerated even after intra-articular injections of hyaluronic acid. The authors are suggested to dig deep and look into the groups (arms) they have made. Otherwise, they should not have expected any change in all four groups when they were giving hyaluronic acid to subjects in all four groups.

Author Response

  • The full form of IRB must be given for clarity (line 96), as in some institutions, it is called IEC (Institutional Ethical Committee), and in some, it is called ERB (Ethics Review Board) or REB (Research ethics board).

Thank you for the suggestion, we have corrected the mention of IRB (Line 102)

  • The authors have written in the next line, ‘informed consent was sought……Declaration’. It should be,’ informed consent was taken. The authors are suggested to change it accordingly.

Thank you for pointing this out, we have corrected accordingly (Line 103)

  • Section 2.3 'Multiple Harvest Method' should be placed before section 2.2 'Treatment'.

As suggested by the Reviewer, we have restructured these sections (line 131 and 143).

  • Number of patients must be mentioned in the section 2. ‘Materials and Methods’ along with the information, how many subjects had bilateral knee OA.

Thank you for the input. In terms of subject recruitment, in section 2, we focused on the methodology of how we acquire subjects, while in the section 3 we presented the results of our recruitment methodology, including number of subjects. Additional information regarding how many subjects had bilateral knee OA has been added (line 219-220).

  • Patients doing regular exercise must be taken in the exclusion criteria, as exercise significantly improves VAS and WOMAC scores.

Thank you for the input. While we appreciate the reviewer’s feedback, we do not put regular exercise as our exclusion criteria, as it has a very wide definition and in certain cases might worsen patient condition if not correctly practiced. In general, we recommended our patient to avoid high impact activity/ exercise during study.

  • In the 4thparagraph (lines 262-271), the authors discussed possible scenarios of why cartilage was not regenerated even after intra-articular injections of hyaluronic acid. The authors are suggested to dig deep and look into the groups (arms) they have made. Otherwise, they should not have expected any change in all four groups when they were giving hyaluronic acid to subjects in all four groups.

In this study, the hyaluronic acid group was only served as control because it is one of the established regular treatment for OA. The rationale of adding growth hormone has been discussed in the 3th paragraph. Several hUC-MSCs groups were created to assess the efficacy of additional growth hormone to the hUC-MSCs implantation, since previous research resulted in better clinical function and increase in cartilage volume as an indicator of regeneration.

(Dilogo IH, Canintika AF, Hanitya AL, Pawitan JA, Liem IK, Pandelaki J. Umbilical cord-derived mesenchymal stem cells for treating osteoarthritis of the knee: a single-arm, open-label study. Eur J Orthop Surg Traumatol [Internet]. 2020;30(5):799–807. Available from: https://doi.org/10.1007/s00590-020-02630-5)

This manuscript is a resubmission of an earlier submission. The following is a list of the peer review reports and author responses from that submission.

Round 1

Reviewer 1 Report

Comments and Suggestions for Authors

This manuscript aim to evaluate the efficacy and safety concern for the combination of hyaluronic acid (HA), human umbilical cord mesenchymal 20 stem cells (hUC-MSCs), and synthetic human growth hormone (somatotropin) in the treating knee OA, assessing pain relief, functional improvement, and cartilage regeneration. Results from VAS and WOMAC indicate that all treatment arms show improvements over 12 months while MRI T2 mapping showed no significant cartilage regeneration across the groups. Before being considered for publication, there are certain issues which need to be addressed properly. 

1. The non-significant observation from MRI T2 mapping may suffer from inadequate detection. Addition evaluation with more sensitive detecting methodology should be applied.

2. Sample size with 28 subjects (51 knees) with 12 month observation may not be adequate enough for solid conclusion to derive from.

3. There lacks detailed description in the methodology section for the possible  randomization process and the handling of dropouts or missing data.

4. The format adopted for organizing cited references is not consistent. 

5. The visualization contents embedded in this manuscript are rather plain

Reviewer 2 Report

Comments and Suggestions for Authors

Comments on Manuscript: biomedicines-3174322

Title: Evaluating synergistic effects of hyaluronic acid, human umbilical cord-derived mesenchymal stem cells, and growth hormones in knee osteoarthritis: A multi-arm randomized trials

The authors present their study to evaluate efficacy and safety of combining hyaluronic acid (HA), human umbilical cord mesenchymal stem cells (hUC-MSCs), and synthetic human growth hormone (somatotropin) in the treatment of knee OA, assessing pain relief, functional improvement, and cartilage regeneration.

The paper is substantially well organized and written, however it has various aspects that induce to deep reflections.

Nerve, retinic and auditive cells, as chondrocytes do not regenerate: thus, several sentences in the Abstract and in the whole manuscript should be reformulate: it is unuseful to aim to something that can not happen. Moreover, it is unavoidable to grow and get old, so it is out of consideration the way to preserve everlast tissue from ageing as a purpose for clinical studies.

It is undoubt that NSAIDS, injections and physical/daily life strategies are not more than symptomatic treatments, but knee artrhroplasty is the unique and definitive solution for late stages or specific arthropathy: thus, surgery can not be put together as in lines 204-206.

PTP is not recommended by any scientific society and should not be considered in any way as a treatment, but just as a dead-end attempt of the past years.

Thus, I think that as it is, the paper is considerable for publication only after major revisions, focused for a realistic hypothetic concept of the chronic diseases, that should be surely managed but not fully solved.

Comments on the Quality of English Language

see attachment